# Charting host-microbe co-metabolism in skin aging and application to metagenomics data

Wynand Alkema[1,☯,¤a], Jos Boekhorst[1,☯,¤b], Robyn T. Eijlander[1], Steve Schnittger[2], Fini De Gruyter[1,¤c], Sabina Lukovac[1,¤d], Kurt Schilling[2], Guus A. M. Kortman[1]*

**1** NIZO Food Research B.V., Ede, The Netherlands, **2** Estée Lauder Companies, Melville, New York, United States of America

☯ These authors contributed equally to this work.
¤a Current address: TenWise B.V., Oss, The Netherlands
¤b Current address: Host-Microbe Interactomics Group, Department of Animal Sciences, Wageningen University & Research, Wageningen, The Netherlands
¤c Current address: Section Genome Diagnostics, Department of Genetics, University Medical Center Utrecht, Utrecht, The Netherlands
¤d Current address: Hub Organoid Technology (HUB), Utrecht, The Netherlands
* Guus.Kortman@nizo.com

**Data Availability Statement:** All relevant data are within the manuscript and its Supporting Information files. In addition, raw 16S amplicon sequencing data has been made available in the

## Abstract

During aging of human skin, a number of intrinsic and extrinsic factors cause the alteration of the skin's structure, function and cutaneous physiology. Many studies have investigated the influence of the skin microbiome on these alterations, but the molecular mechanisms that dictate the interplay between these factors and the skin microbiome are still not fully understood. To obtain more insight into the connection between the skin microbiome and the human physiological processes involved in skin aging, we performed a systematic study on interconnected pathways of human and bacterial metabolic processes that are known to play a role in skin aging. The bacterial genes in these pathways were subsequently used to create Hidden Markov Models (HMMs), which were applied to screen for presence of defined functionalities in both genomic and metagenomic datasets of skin-associated bacteria. These models were further applied on 16S rRNA gene sequencing data from skin microbiota samples derived from female volunteers of two different age groups (25–28 years ('young') and 59–68 years ('old')). The results show that the main bacterial pathways associated with aging skin are those involved in the production of pigmentation intermediates, fatty acids and ceramides. This study furthermore provides evidence for a relation between skin aging and bacterial enzymes involved in protein glycation. Taken together, the results and insights described in this paper provide new leads for intervening with bacterial processes that are associated with aging of human skin.

## Introduction

In the past decade, there has been increased appreciation of the influence of the host microbiome composition and functionalities on human cellular processes, such as gut and skin health and immunity, but also visible aging of the skin. During aging, the human skin changes

European Nucleotide Archive (ENA) with accession number PRJEB45035. HMMs (in .txt files format) are in the supplementary files, and have also been made available in a public Github repository https://github.com/andreiprodan/mask-publication, along with source code and a workflow description that are also available for download.

**Funding:** Estée Lauder Companies is a company selling skin care products, which partly funded this study and provided the samples for this study. The funder provided support in the form of salaries for authors JB, WA, RTE, SL, FdG and GAMK, but did not have any additional role in the study design for data analysis and interpretation, in data collection and analysis and in preparation of the manuscript. The funder supported the decision to publish the study through co-authors KS and SS. The specific roles of these authors are articulated in the 'author contributions' section. TenWise B.V. played no role in funding or decision to publish and is merely mentioned as a current address of one of the NIZO-affiliated co-authors (WA). There are no competing interests with TenWise.

**Competing interests:** At the time of this study, KS and SS were employed by Estée Lauder Companies, a company that sells skin care products and hired NIZO (a contract research organization for (a.o.) microbiota studies) through authors WA, JB, RTE, SL, FdG and GAMK. This does not alter our adherence to PLOS ONE policies on sharing data and materials. There are no further competing interests.

both macroscopically and microscopically. Wrinkle formation, reduced elasticity and wound healing, as well as a reduced barrier function are hallmarks of this process. These phenotypic changes can be caused by intrinsic factors, such as pigmentation, anatomical variations and hormonal changes, and extrinsic factors, such as temperature, life style and exposure to smoke and sunlight. The molecular processes in human cells that are triggered by these signals range from immune related processes to molecular pathways that affect the structural integrity of the skin and the potential for rejuvenation [reviewed in 1].

The human skin is inhabited by a large number and variety of microorganisms, including bacteria, fungi and viruses. The impact of these organisms on skin health and skin aging becomes more and more evident with the maturation of 16S rRNA gene and metagenomics sequencing techniques that allow for the assessment of changes in the skin microbiome upon aging, onset of disease or therapeutic intervention. Recently, a number of studies have investigated the changes in the skin microbiome associated with skin aging and therapeutic intervention [2–14]. In general, the findings of these studies showed that the skin microbiome consists of a number of microbial genera that are consistently found in all humans (notably species from *Staphylococcus*, *Cutibacterium*, *Corynebacterium* and *Acinetobacter*) but that the exact composition is influenced by, amongst others, body site, gender, geographic location and age. In the study by Kim et al (2019), an overrepresentation of *Alistipes*, *Prevotella*, *Porphyromonas*, *Sphingobacterium*, *Lactobacillus*, *Aerococcus*, *Oscillospira* and *Ruminococcus* was found in the younger group (25–35 years old) of healthy female volunteers and an overrepresentation of *Micrococcus*, *Corynebacterium*, *Dermacoccus*, *Actinomyces Streptococcus*, *Lysinibacillus* and *Bacillus* in the older group (56–63 years old) [4]. Dimitriu and co-workers described an increase in *Corynebacterium*, *Neisseriaceae*, *Chryseobacterium*, *Prevotella*, *Veillonella* in older skin and also showed a direct correlation of a number of representatives of these bacterial families with wrinkles and the number of pigmented spots in skin [3]. Similarly, Juge and coworkers described higher levels of *Corynebacterium* and reduced levels of *Propionibacterium* in older skin (54–69 years old) [9].

Although over- and underrepresented bacterial taxa can be readily associated with younger or older skin, the limiting step and challenge in the analysis of these data is the interpretation of biological relevance in context of the research question or hypothesis. Specifically, little insight is currently available in literature on the interplay between microbial functionalities and human cellular processes (referred to as co-metabolism). In order to do this correctly, profound knowledge of the physiology of bacterial skin residents needs to be combined with knowledge on the physiology of the host that is related to the skin condition of interest.

Currently, analysis of such multidimensional data is performed by mapping to canonical knowledge that is stored in databases, for instance to determine enriched biological processes that are modified upon intervention or during the progression of a disease [15]. Whereas this is a valid approach, the value of the outcome is intrinsically limited by the lack of databases that store cross-domain knowledge on co-metabolism between microbiological and human cellular processes (e.g. KEGG and GMM) [16–18].

To increase our understanding of the influence of microbial functionalities (represented by the skin microbiome) on molecular processes involved in aging of human skin, there is a need for improved computational data analysis workflows that allow for integration of knowledge and data on both aspects. In this paper, we describe a conceptual framework to substantiate functional links between the human skin microbiome (cheek area) and cellular processes involved in age-related skin appearance. Similar workflows have previously been successfully employed [19, 20].

First, we constructed an expert-curated set of bacterial functionalities (genes) linked to skin aging-related intrinsic processes that was obtained through an extensive literature study. We

applied bioinformatics approaches to identify these genes in publicly available bacterial genome sequences. To substantiate literature findings and confirm the link between specific microbial taxa to aging skin, skin microbiota compositional data was generated through 16S rRNA gene amplicon sequencing of microbial DNA isolated from cheek skin samples of female subjects with different age-related skin appearance. The functional potential of the defined microbial communities was inferred by combining compositional data with gene content information from reference genomes of skin bacteria. Finally, the described methodology was validated on a publicly available skin-related shotgun metagenomics dataset.

## Materials and methods

### Literature searches

Literature searches were carried out in June 2019 by running multiple searches on MedLine using the following search term " "skin aging" OR "skin rejuvenation" OR (skin AND "extrinsic aging") OR (skin AND "intrinsic aging") ". This yielded 8616 abstracts that were subsequently evaluated. Evaluation was done by manually screening the abstracts for statements on molecular pathways, metabolites, microorganisms and microbial genes. For a selected number of abstracts containing such statements, full text papers were collected and evaluated. From these full text papers, citations were screened and, where relevant and not yet retrieved in the first search, added to the set of references. Finally, from this set of literature references, gene symbols and/or locus tags encoding microbial functionalities were retrieved to be used as input for the generation of Hidden Markov Models (HMMs) (see below).

### Hidden Markov Models

**Reference genomes.** For the relevant microbial functionalities that were retrieved from literature, reference genomes were collected from the NCBI sequence repository. Selection was based on availability of genome sequences of specific strains mentioned in the literature references. If the genome sequence of the strain mentioned was not available, a sequence from a representative strain from the same species was included in the reference genome set as an alternative. Alternative strains for which the source of isolation was marked as 'skin' were given priority over strains from a non-skin related or unknown origin. The final list of reference genomes is provided in S1 Table. A maximum-likelihood phylogenetic tree of selected reference genomes was generated with FastTree [21], based on concatenated multiple sequence alignment of all genes with exactly one copy on each of the genomes, as determined with OrthAgogue [22].

**Generation of Hidden Markov Models.** Bacterial functionalities with a common evolutionary origin in the reference genome set (see section 'Reference genomes') were identified using OrthAgogue. Multiple sequence alignments were generated with Muscle [23] for each orthologous group containing one or more genes deemed relevant for skin aging processes, as described under "Literature searches". From these multiple sequence alignments, HMMs were constructed using *hmmbuild* (https://hmmer.org). The HMMs were designed specifically for skin microorganisms (in contrast to already existing signatures in publicly available databases that are based on a generic and broad range of microorganisms), which limits the application of these models to this specific application. For each model, a score threshold (GA) was determined through a heuristic approach and manual curation by taking the average of the lowest-scoring true positive and the highest-scoring true negative, using all proteins from the orthologous group as true positives and all other proteins from the reference genome set as true negatives. These scores were included in the individual HMM files as gathering threshold (GA). A

workflow description for generating the HMMs, including the python script used for this study, are provided in a Github repository.

**Relative abundance of bacterial functionalities associated with skin aging.** To assess the abundance of bacterial genes in the identified pathways through application of the generated HMM's (see section 'Generation of Hidden Markov Models'), publicly available metagenomics datasets of a skin-derived and gut-derived microbial population was collected from literature [24, 25]. From these studies, samples were chosen based on the provided metadata of the subjects and/or sampling site that were included in these studies. The skin samples were chosen based on healthy volunteers collected through a similar sampling technique (swabs) on a similar sampling site (facial skin) as applied in this study (see section below). The gut samples were chosen based on healthy individuals who did not recently use antibiotics. Assembly of these reads was done using *Spades* (version 3.11.1) with the "—meta" flag enabled. Genes were called using *Prodigal* (version 2.6.2) [26]. Read mapping was performed by first making a nucleotide database using Bowtie2 (version 2.2.3) and then mapping the reads to the called genes using *HUMANn2* (version 0.9.9) with the "—bypass-nucleotide-index", "—bypass-translated-search" and "—nucleotide-database" flags enabled. Bacterial functionalities associated with cellular processes involved in skin aging were identified in the bacterial genomes and shotgun metagenomics datasets by scanning protein-coding sequences (either as provided genome annotations, or as described above) using *hmmsearch* (version 3.1b2) (https://hmmer.org) with HMMs and score thresholds as described above. Overview tables with per-sample counts (either of genome sequences or shotgun metagenomics samples) of each HMM were generated using Python (https://python.org) and HUMANn2. Model level counts were generated by summing up the counts for the individual members in the model. By using two additional HUMANn2 functions (humann2_join_tables and humann2_renorm_table) the relative abundance was calculated between all samples, to diminish the differences in read number between samples and between the two different datasets. Finally, the relative abundance was normalized for each individual module by dividing all gut and skin samples within one module by the sample with the highest value. Through this approach, all samples within one module are relative to each other, between 0 and 1.

## Sample collection and phenotypic assessment

To substantiate literature findings, skin microbiota compositional data was generated through 16S rRNA gene sequencing of microbial DNA isolated from skin swab samples (cheek area) of 2 x 25 healthy female volunteers of two different age groups (20–28 years old ('young') and 59–68 years old, ('old')).

**Study design and skin swab collection.** A human trial was performed for single sample collection of cheek samples from two groups of female subjects from the general European descent population (n = 25 per group) in Belgium. The subject age was between 20 and 28 years (younger population group), and between 59 and 68 years (older population group). Exclusion criteria were: nodulo-cystic lesions/acne or sebaceous gland condition; eczema, psoriasis, atopy; prescribed or unprescribed use of skin treatment within 1 month prior to inclusion (oral or topical antibiotic, antifungal or topical steroids); smokers or having smoked in the past 2 years; recent history of chronic alcohol consumption defined as more than 15 standard servings per week or more than 3 servings per day; tanning bed usage less than 1 month prior to inclusion; sunbathing 1 month prior to inclusion; habitual exposure to sun or use of a tanning bed; clear UV light effects in the younger subjects that might be related to aging (inspected by the principal investigator during the recruitment procedure); primary immunodeficiency patients known to have dysbiosis in community diversity; use of tanning

dihydroxyacetone (DHA) less than 1 month prior to study start; excessive habitual caffeine use (more than 6 small/medium cups of coffee or soda daily); body mass index higher than 30; pregnancy or lactation. At the moment of inclusion, subjects were requested to: refrain from using OTC products for any kind of skin treatment 7 days prior to the sampling; refrain from any facial creams and make-up on the day of the sampling; refrain from swimming in a chlorinated pool, using a hot tub, sauna/steam baths, 48 hours prior to sampling visit; refrain from any facial treatments, facial masks, scrubs/peelings 2 weeks prior to the sampling visit; to follow specific bath/shower procedures (showering with plain water only, soap should be avoided as much as possible, and no scrubbing of the skin with a towel) during 24 hours prior to sampling visit–on the day of sampling showering was avoided completely (also with plain water).

The study was conducted according to the principles of the Declaration of Helsinki latest version Fortaleza, Brazil, October 2013. The candidates were informed verbally on the aim of the study and the study procedures. All participating subject signed the informed consent form. Data was analyzed anonymously. No IRB approval was requested for this study, for two reasons. Firstly, the study was carried out in Belgium, where IRB approval for this type of studies is not required. Secondly, there was no intervention or invasive procedure involved for the collection of the skin swab samples. Participants were requested to fill out a questionnaire (not considered as a psychological burden), and were only subjected to mild skin characteristics measurements.

During site visit information on the following parameters/metadata was collected: skin hydration (Corneometer, Courage&Khazaka); transepidermal water loss (TEWL) (Aquaflux, Biox); skin pH (Metrhom); skin smoothness, scaliness, sebum spots (Visioscan, Courage&Khazaka); standardized images were taken from the face (frontal and left view) with the Visia-CR (Canfield) to assess invisible and visible spots, red features, pores, porphyrins, color, fine lines, wrinkles and roughness. Additional metadata on skin properties was collected during the study by means of a questionnaire.

Samples were collected by means of swab sampling from the intact cheek skin. In short, skin sampling was performed using a standard custom-made sampling template, which allows for consistent sampling of 2 cm$^2$ of the cheek areas (both sides used). Before sampling of each subject, the template was cleaned with ethanol (70%) and air dried. To further minimize sample cross-contamination, a fresh pair of sterile gloves was worn by the person sampling each individual. The area within the template was swabbed with a sterile HydraFlock collection swab (3206H-25; Puritan Diagnostics, USA). The collection swab was soaked in sterile PBS + solution (PBS of pH 7.0 with 0.5% Tween-20). Next, the swab was dried very briefly on a sterile gauze before the start of the sampling. Samples were taken by direct swabbing of the cheek skin. The shaft of the swab was held parallel to the skin surface and it was rubbed back and forth 10 times applying firm pressure. Immediately after swabbing, each swab was swirled in a 1.5 ml collection tube with 0.25 ml of sterile PBS+ solution with 2.5 µl of 1 mM EDTA (DNase inhibitor). The swab was left in the solution and the samples were stored on ice before freezing and shipment on dry ice for further processing at NIZO (The Netherlands).

**16S rRNA gene sequencing and bioinformatics.** A detailed description of the DNA isolation, 16S rRNA gene sequencing, bioinformatics and statistical methods is provided in S1 Appendix. In short, sequencing data was analysed with a bioinformatics workflow based on Qiime 1.8 [27] and functional potential of the bacterial communities was inferred using a method analogous to PICRUSt [28]. Correlation of specific microbial taxa to skin age was primarily based on the skin aging score of the samples collected. This skin aging score was determined using eight skin age-related physiological measurements and was calculated as described in S1 Appendix. Raw sequencing data is available from the European Nucleotide Archive (ENA) with accession number PRJEB45035.

## Results

To define the influence of microbial functionalities represented by the skin microbiome on molecular processes involved in aging of human skin, a multistep bioinformatics and data analysis approach was applied (Fig 1). First, scientific literature was evaluated to identify shared metabolic biological pathways involved in skin aging between humans and skin micro-organisms. Based on the microbial pathways, Hidden Markov Models (HMMs) were constructed that were subsequently used to analyze the reference genomes from skin organisms, metagenomic data and 16S rRNA gene profiling data. The details of each step are described in the Materials and Methods section. Together, conceptual framework substantiates functional links between the human skin microbiome (cheek area) and cellular processes involved in age-related skin appearance as described in the sections below.

### Co-metabolic processes involved in skin aging

Through an extensive literature study, we selected molecular processes in human cells that are involved in skin aging for which associations with microbial functionalities in the skin microbiome were described. Host pathways were designated after the literature search and were used for a targeted investigation of samples from subjects included in this study. An overview is provided in Table 1 and detailed descriptions of each process and co-metabolism pathways involved are provided in S2 Appendix.

The pathways listed in Table 1 are centred around a single metabolite (e.g. urocanic acid), entire pathways (e.g. histidine conversion, ceramide biosynthesis) or a more general group of

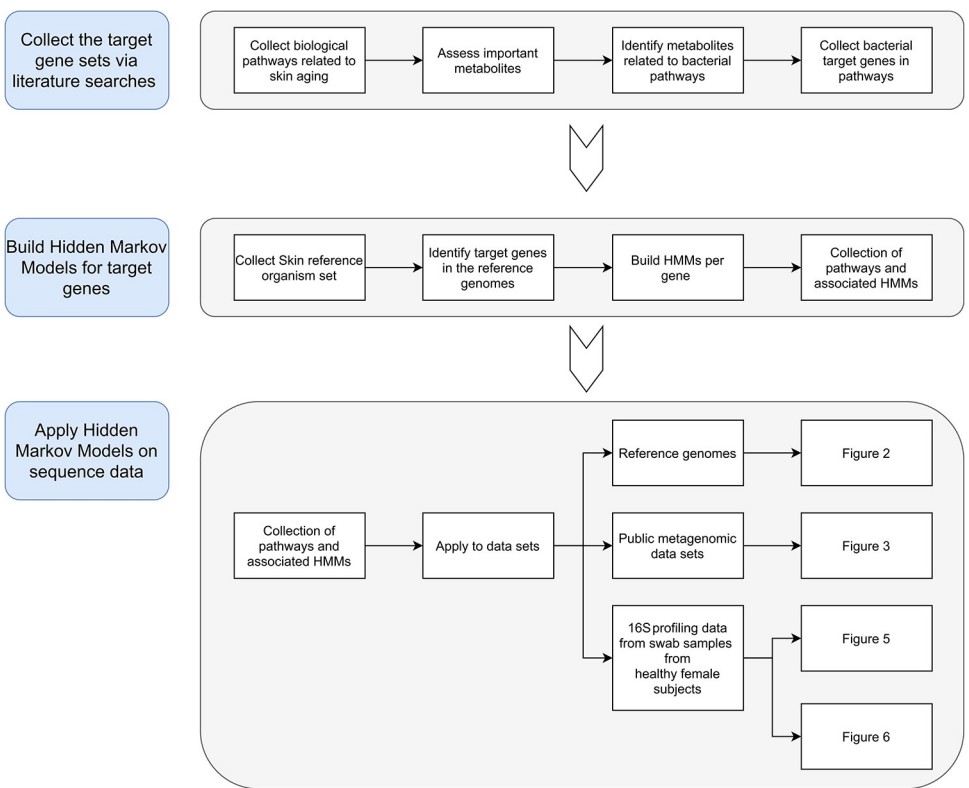

**Fig 1. Schematic overview of the workflow applied this study.** The description of the multistep approach is linked to the results obtained and described in this manuscript (figure numbers).

**Table 1. Co-metabolic process involved in skin aging.**

| Host Process | Rationale | Targets for co-metabolism | Bacterial genes/ functionalities |
|---|---|---|---|
| **UV-B induced immune suppression** | UVB radiation causes immune suppression, mediated by urocanic acid. | urocanic acid | *hutH* |
| **Histidine conversion** | Histidine and other amino acids act as natural moisturizers on the skin and display antimicrobial activity. | histidine, other amino acids | *hut*-operon |
| **Protein glycation** | Glycation of collagen type 1 and other structural proteins is a major cause of loss of skin elasticity. | collagen, vimentin, elastin | fructokinases |
| **Pigmentation** | Production of melanin via the conversion of tyrosine and phenylalanine is a major pathway in pigmentation. | tyrosine, phenylalanine | tyrosinase |
| **Ceramide metabolism** | Incorrect ceramide metabolism has been associated with a decreased barrier function. | ceramides, sphingolipids, fatty acids | ceramide metabolism (*cerN*, *sphR*). |
| **Fatty acid metabolism** | Fatty acids are of major importance as carbon source, antimicrobial compounds and signalling molecules. | Fatty acids, in particular long chain fatty acids | Fatty acid metabolism, fatty acid biosynthesis, beta-oxidation, *Fad* operon. |
| **Lipoteichoic acid signaling** | LTA interacts with TLR2 and influences the recruitment of immune cells. | lipoteichoic acids | LTA biosynthesis (*tag* operon, LtaS) |
| **Porphyrin synthesis** | Increased porphyrin levels are often used as an indicator of skin aging. | porphyrins, heme | porphyrin biosynthesis, *deoR* Hem-operon |
| **Proteolysis** | Bacterial proteases can degrade structural proteins in the skin | collagen, elastin | *lasA*, *lasB*, *sspA*, *sspB*, *sspC* |
| **Oxygen radical production and scavenging** | Increased production of ROS and reduced capacity to scavenge these radicals. | oxygen radicals, glutathione | catalase, superoxide dismutase |

Most important host processes involved in skin aging, including targets for assumed microbiome co-metabolism and associated bacterial genes or functionalities, as further described in S2 Appendix. The targets for co-metabolism can either be metabolites that are produced by human or bacterial cells, or human proteins (e.g. receptors) that can be targeted by the metabolites.

molecules (e.g. fructokinases that may play a role in glycation of multiple distinct proteins). It should be noted that this table is not a comprehensive reflection of the entire spectrum of metabolic pathways that play a role in bacterial-host interaction. For example, it is well known that skin bacteria interact with the host immune system by activation of Toll like receptors (TLRs) [29]. However, some important ligands for these TLRs, such as dsDNA, ssDNA or cell wall fragments, cannot be linked to specific bacterial genes or operons and are therefore not included in this overview.

## Genomic distribution of genes involved in skin aging

To study the distribution of microbial functionalities defined in Table 1 across a set of relevant skin microbiota-specific microorganisms, genome sequences of such bacteria were obtained from NCBI as described in the Materials and Methods section (S1 Table). The proteins encoded by these genomes were scanned with HMMs representing the aging-related functionalities described above. For each organism the number of genes predicted to encode these functions was identified (Fig 2).

This analysis shows that subsets of related species are responsible for specific functionalities of interest, instead of equal distribution of the functionalities across the species. For example, species from the *Anaerococcus* and *Lactobacillus* group do in general score low on all the modules; they do not contain the full pathways for porphyrins metabolism or that code for proteins that are involved in proteolysis on the skin. This is in line with the fact that porphyrin metabolism has not been described in literature for species of these groups. In contrast, the porphyrins metabolism pathways are fully represented in species of *Corynebacterium* and *Cutibacterium* (formerly known as *Propionibacterium*) species (Fig 2). Another notable feature is the relative lack of genes for fatty-acid beta oxidation in *Staphylococcus* species, which is in agreement

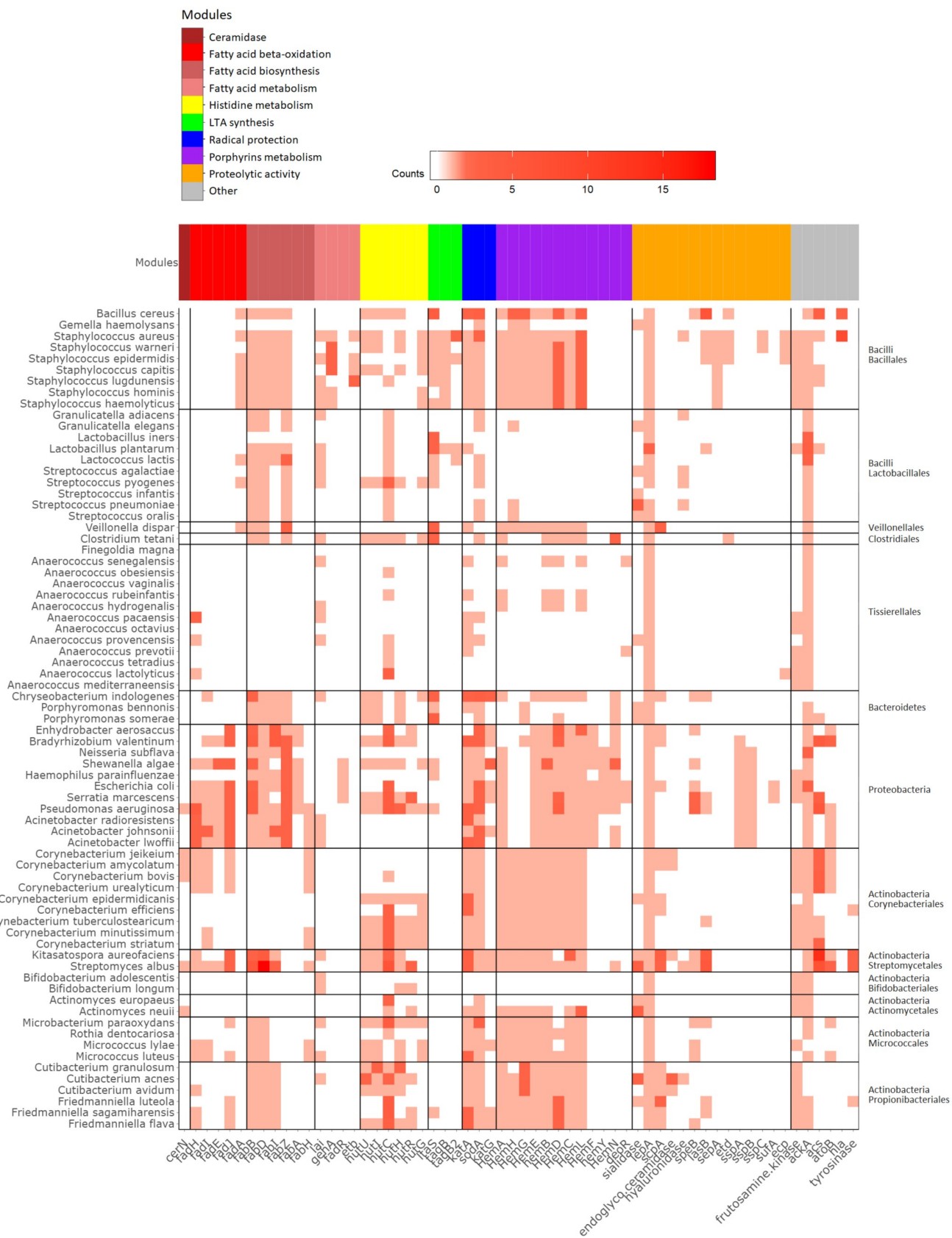

**Fig 2. Skin aging module scores in reference genomes.** Different functionalities are shown in columns, the reference genomes are shown in rows. The columns are clustered based on modules within a known pathway (color-coded) and the individual modules from that pathway are shown on the x-axis. The number of genes predicted to encode the specified functions is shown on a scale from white to red, with red showing the highest number.

with the large differences previously described for fatty acid uptake and metabolism between Gram-negative and Gram-positive organisms and among Gram-positive organisms [30].

The illustration in Fig 2 highlights the fact that the skin microbiome as a whole performs functions that cannot be performed by a single species and that complex cross-interactions play a crucial role in understanding host-related processes.

## Metagenomic distribution of genes involved in skin aging

To assess the abundance of the identified bacterial genes in the pathways, the HMMs were also applied on metagenomic datasets derived from gut [24] and skin samples [25] (Fig 3). The most notable differences observed between both dataset types are those in the modules related to histidine metabolism, porphyrin metabolism and proteolysis. Importantly, none of the modules is consistently fully covered within a single sample, and especially the fatty acid beta-oxidation is not well represented on the skin. This implicates that not all functionalities are yet fully presented in the current collection of HMMs.

## Distribution of genes in aging skin

To define bacterial microorganisms that are associated with younger or aging skin, we generated a microbiota compositional data set based on 16S rRNA gene sequencing obtained from microbial DNA isolated from skin samples (superficial layer, cheek area) from healthy, female subjects of two different age groups, as described in S1 Appendix. Analysis of this dataset shows a significant link between samples of subjects from different age groups ('young' and 'old') and bacterial composition on the OTU (Operational Taxonomic Unit) level (Fig 4). Inspection of the redundancy analysis (RDA) for variation in microbiome composition between the samples of both age groups, shows that, amongst others, *Corynebacterium*, *Acinetobacter*, *Leptotrichia*, *Veillonella* and *Chryseobacterium* associate with older skin.

To further quantify the correlation between microbiome composition and skin age, a skin aging (SA) score was calculated for each sample based on eight clinical measurements that represent phenotypic appearance of aging skin (namely pores number, roughness, wrinkles number, porphyrins number, red features number, skin color evenness, spots-visible number, and spots-invisible number) as described in S1 Appendix. In short, a high SA score means that there is a high score on the above parameters that are in general associated with an aging skin. This approach is similar to the methodology used by Dimitru and co-workers [3], with the exception that in the SA score more parameters are included. Fig 5A shows the link between the SA score and the microbiome composition for skin swab samples from female volunteers of the 'young' age group. Interestingly, this analysis already shows a significant link between the microbiota composition and SA scores. We only included the samples of the 'young' group in this analysis in order to remove the large effect of chronological age on the microbiome (Fig 3). This analysis thus shows a relation with the microbiome that is purely based on actual (age-related) appearance of the skin and not the defined chronological age. Interestingly, a higher SA score in this subject group is mainly driven by *Propionibacterium* and not by the organisms that were shown to be associated with chronological age (Fig 4).

Next, the functional potential of the defined skin microbiota was predicted by combining data on the presence and absence of bacterial genes linked to skin aging processes using the compositional profiles derived from the 16S rRNA gene sequence data. As shown in Fig 5B, a

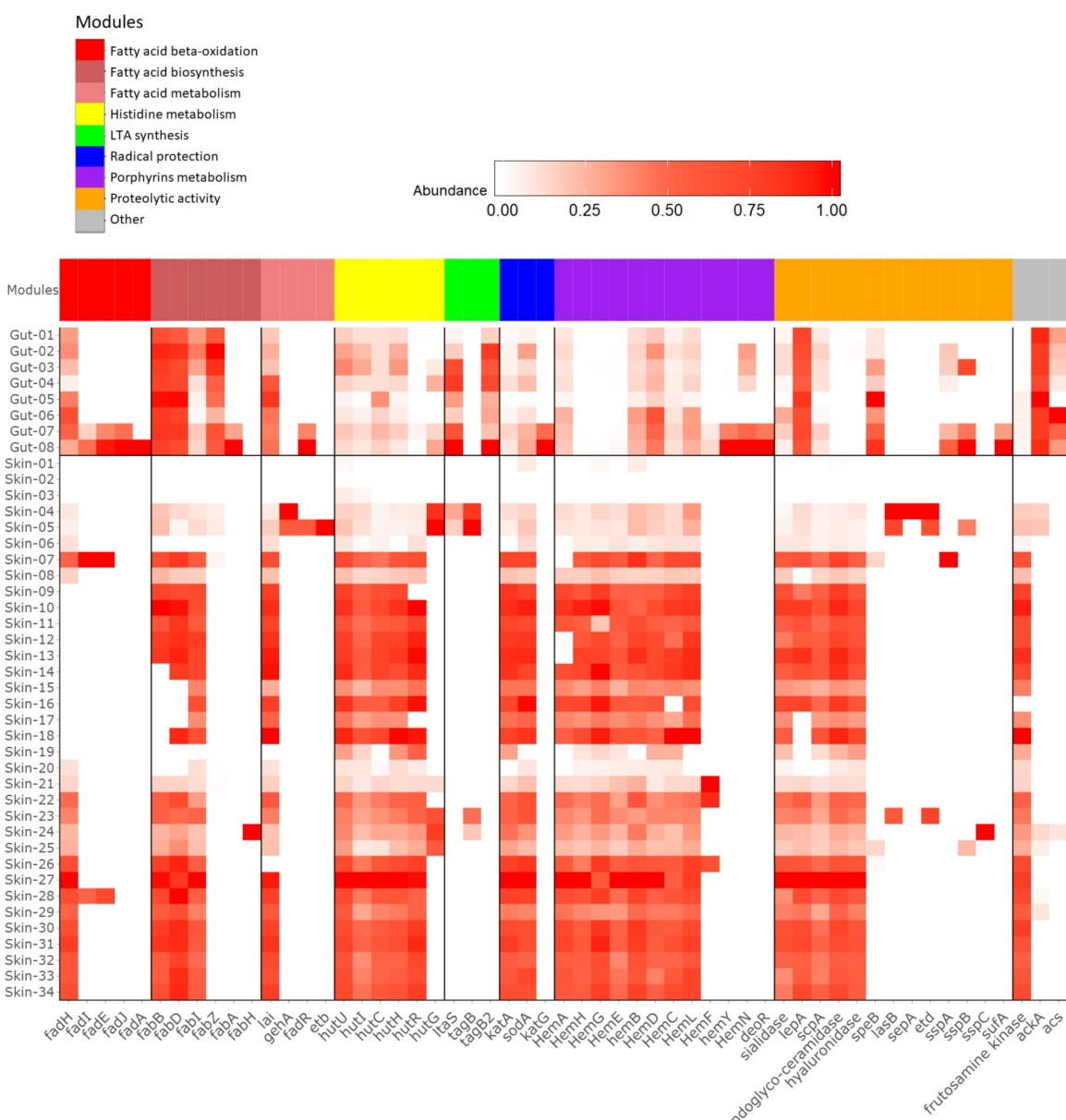

**Fig 3. Hits for the HMM modules on metagenomic datasets from gut and skin samples.** Different functionalities are shown in columns whereas the publicly available metagenomic datasets from gut and skin samples are shown in rows. The columns are clustered based on modules within a known pathway (color-coded) and the individual modules from that pathway are shown on the x-axis. The relative abundance of each functionality is shown on a scale from white to red, with red showing the highest level of abundance. Abundance was normalized to values between 0 and 1 within each module (columns) by dividing all gut and skin samples within one module by the sample with the highest value.

number of genes are significantly linked to the SA score, These genes mostly belong to i) the fatty acid beta-oxidation pathway (*fadI*, *fadJ* and *fadB* genes) and ceramide pathway (*cerN*), which are negatively correlated to the SA score, ii) the genes related to fatty acid biosynthesis (*fabI*, *fabH* and *fabG* genes) and iii) genes from histidine metabolism (*hutH*, *hutU* and *hutL*), which are positively correlated to the SA score. In addition, genes involved in response to oxidative stress (*sodA*, *katA*, *katG*) are related to the SA score, but the direction of this response is

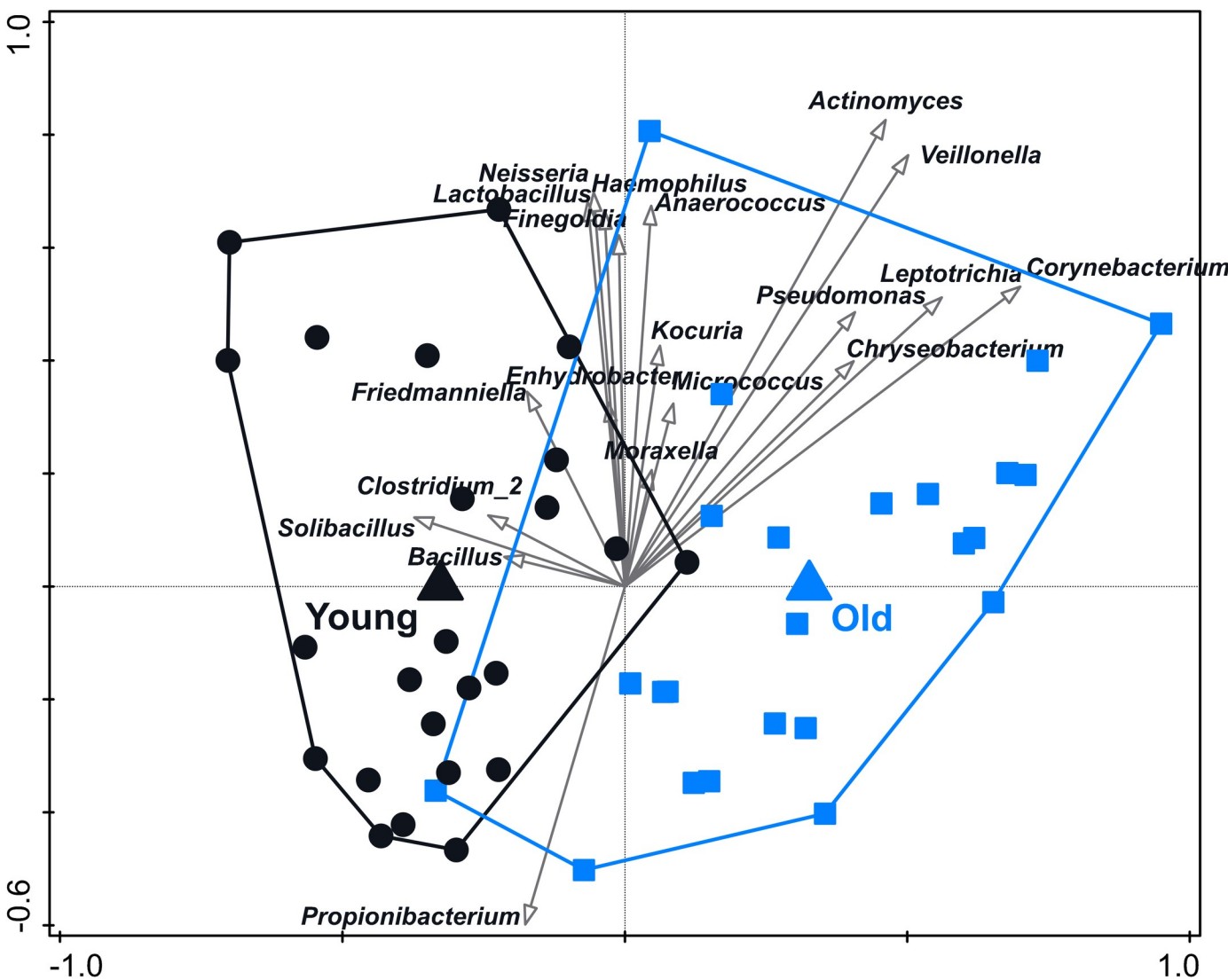

**Fig 4. Variation in microbiome composition of cheek skin samples from female volunteers of 'young' and 'old' age groups, based on OTUs.** Samples were taken from the superficial layer of the cheek. Age group ('young' and 'old') explains 1.1% of the variation in the microbiome. Separation of samples by age group was significant (p = 0.044). Samples from the 'young' and 'old' age group are indicated by black circles and blue squares, respectively. Arrows are plotted supplementary and represent 20 bacterial genera that are associated most with the age groups. The length and direction of the arrows indicate the relative strength of the association with either group.

ambiguous; *katG* is associated with low SA score and the functionally overlapping *katA* is associated with a higher SA score.

Human pathways related to skin aging were identified for which metabolites were also present in molecular pathways of skin commensals. However, this does not necessarily imply that all bacterial pathways are also operational on the skin, or active in such a way that they actually influence the process of skin aging. Therefore, based on the counts for the individual genes of all skin aging associated pathways, a pathway score was calculated by adding up all the individual counts for each gene in a pathway. These pathway scores were subsequently compared to the SA score for the individual subjects by creating a linear model for each pathway (Fig 6 and S2 Table). This analysis revealed that only in the 'young' subjects there was a significant correlation between a number of pathways and the skin aging score of these subjects. For subjects in

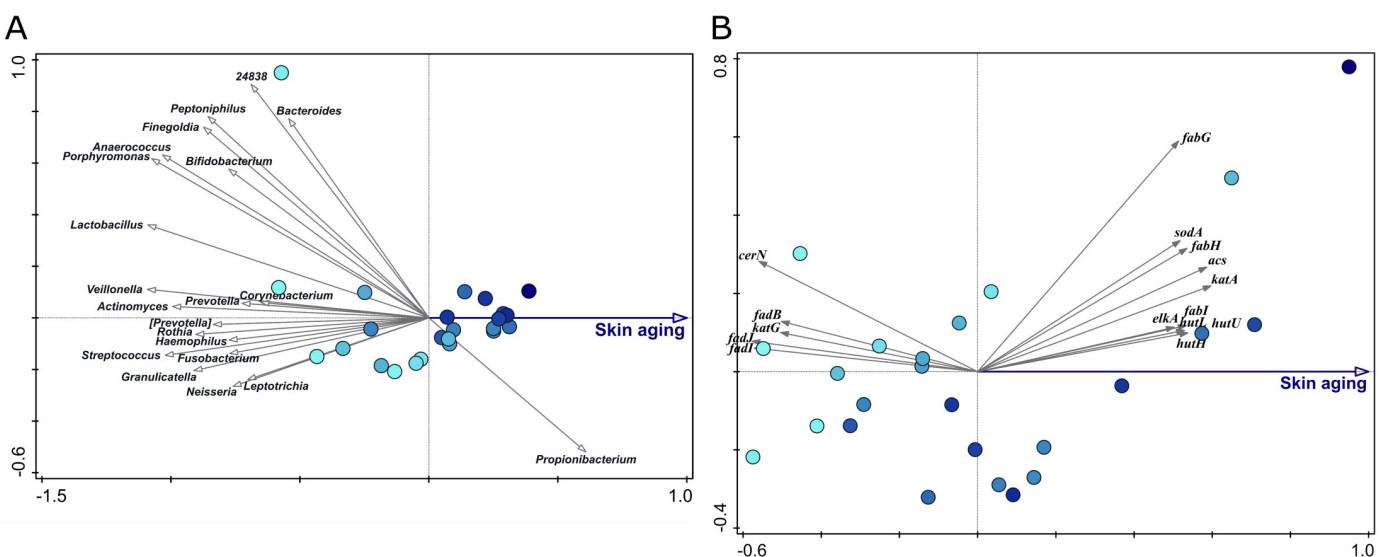

**Fig 5. RDA at the OTU and gene level for all cheek samples derived from subjects of the young age group. A]** RDA on OTU level. The SA score explains 6% of the variation in the microbiome. Separation of samples by SA score was statistically significant (p = 0.002). Blue gradient of the sample symbols indicates relative SA score value (dark blue = high SA score). **B]** RDA on microbial genes involved in skin aging processes for all cheek samples (superficial layer) derived from subjects of the 'young' age group. The SA score explains 7.7% of the variation in the predicted skin aging-related genes of the microbiome. Separation of samples by SA score was statistically significant (p = 0.026). Grey arrows represent the 15 predicted genes that show the highest association with low or high SA score.

the 'old' age group, such a relation was not found (S2 Table). The microbial pathways for radical protection, histidine metabolism and fatty acid biosynthesis are significantly more prevalent in skin with a high SA score (i.e. older looking skin of young female subjects, S2 Table).

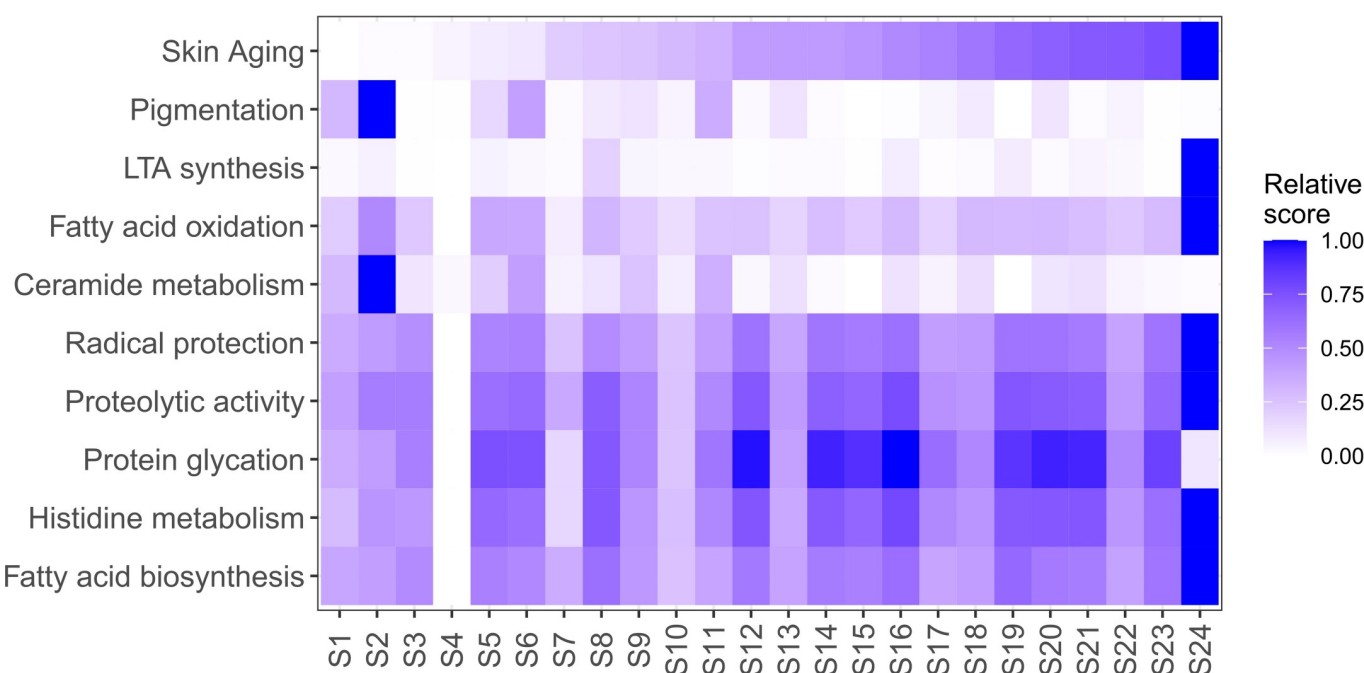

**Fig 6. Normalized functional pathway level scores for subjects of the young age group.** All subjects are in columns and were ordered based on SA score, in which a low SA score represents a young-looking skin and a high score corresponds to older-looking skin (from left to right). Functional pathways involved in skin aging (derived from Table 1) are in rows and were normalized to the maximum score per row.

This higher level pathway integration is in agreement with the redundancy plots in Fig 5 that show that the abundance of a number of the individual genes from these pathways were associated with the SA score.

## Discussion

This study focused on an approach for biological interpretation of 16S rRNA gene and metagenomic profiling data in relation to the intrinsic and extrinsic processes that define skin aging. This approach is an addition to the large number of tools that currently exist for the analysis of community sequence data, including methods for sequence read mapping, statistical comparisons of multi-dimensional data and functional inference [reviewed in 31]. Our approach addresses the challenge of translating a list of OTUs and associated taxa to a description of functional pathways and molecules that can shed light on the actual mode of action of skin aging processes. Although we have exemplified its use on bacterial genomes, 16S rRNA gene and bacterial community sequencing data, the HMMs presented in this paper can also directly be used on metatranscriptomic data. The use of HMMs in sequence-based approaches has been used before [19, 20] but the construction of a HMM set for multiple skin aging-related pathways is new.

Application of the models described in this manuscript on a set of skin-related reference genomes shows that there is a clade-specific distribution of these functional pathways. Together with the fact that some essential pathways, such as fatty acid conversion, are not found in typical skin organisms such as *S. epidermidis* and *S. warneri*. This finding indicates that the set of modules described in this study is not yet complete and extension towards more clade-specific modules is warranted. The process described in this paper uses a manual process for creating these HMMs based on manual curation of a set of scientific articles. Integrating automated text mining methods [32] and automated HMM construction [33] could facilitate this process.

In this study, in which 16S rRNA gene sequencing data was profiled from a cohort of female subjects of 'young' and 'old' age groups, an association of (amongst others) *Corynebacterium*, *Veillonella* and *Chryseobacterium* with the old age group was found, which is in agreement with earlier reports [3, 4, 9]. However, when the microbiome was correlated with a set of phenotypic read outs aggregated in a skin aging (SA) score based on eight clinical parameters associated with older skin appearance, predominantly a higher relative abundance of *Propionibacterium* was associated with older-looking skin. This underscores the importance of selecting the most relevant phenotypic variable when running the statistical analysis.

When aggregating to a pathway level, the data from this study demonstrate that skin with a higher SA score is associated with an increased presence of bacterial metabolic pathways involved in radical protection and protein glycation, and underrepresentation of genes related to pigmentation and LTA synthesis. Based on these data, we assume that these microbial pathways play a role in the physiological processes for skin aging. However, inferring a causal relationship, or even inferring that these pathways are also functionally more active than other pathways in these groups, can only be justified when more data are available on the actual levels of metabolites and transcripts of these genes. Moreover, a more sophisticated method for scoring could be developed that takes into account the dependence and redundancy of gene functions in pathways, rather than simply counting the number of instances of these genes.

Taken together, the approach described in this study (Fig 1) should be regarded as a useful approach for generation of hypotheses with respect to the involvement of bacterial pathways in human physiological process, and as such generate leads for dedicated follow-up experiments. This approach can also be applied on data derived from other body sites (e.g. gut, oral

cavity, etc), although the set of reference organisms to include should be adapted to reflect relevance to that particular body site. In case of application of the approach on different site on skin, the same reference organisms can be used as described in this manuscript as these are a relevant representation of organisms that can be found at multiple sites on the skin. In all cases, targeted verification in wet-lab studies will be required. These experiments could be directed at measuring the transcript levels of bacterial genes, profiling the metabolite profiles or in depth (intervention) assessment of the effect of host-related processes [34]. Together with an improved and extended collection of HMMs for skin aging-related processes, these experiments will provide the next step in understanding the molecular processes in skin aging and the role of the microbiome in these processes.

## Supporting information

**S1 Table. Reference genome list.** List of reference genomes used for the generation of Hidden Markov Models.
(XLSX)

**S2 Table. Significant relationship between SA scores and pathway level scores in skin swab samples of 'young' and 'old' subjects.**
(DOCX)

**S1 Appendix. Materials and methods for phenotypic assessment.** Detailed description of materials and methods applied for the generation of a 16S rRNA gene dataset on the microbiota of cheek skin samples.
(PDF)

**S2 Appendix. Co-metabolic processes involved in skin aging.** Detailed description of co-metabolic processes in human cells that are involved in skin aging with links to microbial functionalities in the skin microbiome.
(PDF)

**S1 Dataset. Raw data HMM scripts.**
(HMM)

**S2 Dataset. Raw data HMM input file with gene symbols, locus tags, organisms and functions.**
(XLSX)

**S3 Dataset. Raw data file with metadata and relative abundances of OTUs, taxa and predicted functionalities.**
(XLSX)

## Acknowledgments

The authors would like to thank Shannon Sanacora and Andrei Prodan for their contributions to this manuscript.

## Author Contributions

**Conceptualization:** Wynand Alkema, Jos Boekhorst, Sabina Lukovac, Guus A. M. Kortman.

**Data curation:** Wynand Alkema, Jos Boekhorst, Sabina Lukovac, Guus A. M. Kortman.

**Formal analysis:** Wynand Alkema, Jos Boekhorst, Fini De Gruyter, Sabina Lukovac, Guus A. M. Kortman.

**Investigation:** Wynand Alkema, Jos Boekhorst, Sabina Lukovac, Guus A. M. Kortman.

**Methodology:** Wynand Alkema, Jos Boekhorst, Guus A. M. Kortman.

**Project administration:** Robyn T. Eijlander, Sabina Lukovac.

**Supervision:** Robyn T. Eijlander, Sabina Lukovac.

**Validation:** Wynand Alkema, Guus A. M. Kortman.

**Visualization:** Wynand Alkema, Jos Boekhorst, Fini De Gruyter, Guus A. M. Kortman.

**Writing – original draft:** Wynand Alkema, Jos Boekhorst, Fini De Gruyter, Guus A. M. Kortman.

**Writing – review & editing:** Robyn T. Eijlander, Steve Schnittger, Sabina Lukovac, Kurt Schilling, Guus A. M. Kortman.

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
