## [Decision Letter · Decision Letter 0]

25 Mar 2021

PONE-D-20-39781

Charting host-microbe co-metabolism in skin aging and application to metagenomics data

PLOS ONE

Dear Dr. Kortman,

Thank you for submitting your manuscript to PLOS ONE. After careful consideration, we feel that it has merit but does not fully meet PLOS ONE’s publication criteria as it currently stands. Therefore, we invite you to submit a revised version of the manuscript that addresses the points raised during the review process.

Your manuscript has been evaluated by two experts in the field. They both found the study interesting but identified several points for improvement. Hence, I invite you to resubmit a revised version of your manuscript in which all points of the reviewers have been addressed. Of note, please make sure to use the term "16S rRNA gene" consistently. Terms like "16S sequencing" are not correct.

We look forward to receiving your revised manuscript.

Kind regards,

Erwin G Zoetendal, PhD

Academic Editor

PLOS ONE

Journal Requirements:

'The authors have declared that no competing interests exist.'

We note that one or more of the authors are employed by a commercial company: NIZO Food Research B.V,  Estée Lauder Companies & TenWise B.V.

Additional Editor Comments (if provided):

Reviewers' comments:

Reviewer's Responses to Questions

**Comments to the Author**

1. Is the manuscript technically sound, and do the data support the conclusions?

Reviewer #1: Partly

Reviewer #2: No

2. Has the statistical analysis been performed appropriately and rigorously? 

Reviewer #1: Yes

Reviewer #2: No

3. Have the authors made all data underlying the findings in their manuscript fully available?

Reviewer #1: Yes

Reviewer #2: No

4. Is the manuscript presented in an intelligible fashion and written in standard English?

Reviewer #1: Yes

Reviewer #2: Yes

5. Review Comments to the Author

Reviewer #1: This work investigates host-microbe co-metabolism (interconnected functional pathways) in skin aging based on metagenomics data. The work is based on Hidden Markov Models (HMM) that were designed to recognize activity of specific functionally relevant pathways from a combination of (host) genomic and (microbiome) metagenomic data sets on skin-associated bacteria. In addition, the models are applied to young and old adult females using predictions of the metagenomic composition, obtained by combining 16S profiles with sequenced microbial genome information. Specific pathway activity, associated with aging, is reported.

This work is based on a combination of well-established methods from biostatistics, bioinformatics and machine learning. The language and reporting are clear, appropriate citations are provided and I do not observe apparent flaws in the methodology. The work contributes a new algorithmic approach to host-associate microbiome research, where the innovation and benchmarking of such tools is necessary and valuable for the progress of the field. The discussion acknowledges some important limitations, such as the current need for substantial manual input during the method construction.

Main limitations of this work from my (methodological) perspective relate to the level of reporting regarding technical detail, and I comment on these below.

* Major

1) More details on the various steps in data processing and analysis could be added. In particular, a schematic figure (or figures) summarizing the approach and experiments would help to more easily understand the overall study setup and put the various pieces in context. The Methods section should then be compared to such overview in order to make sure that the necessary technical details of each step are provided so that the research is reproducible.

2) I did not find mention of the source code availability. It is paramount that the code for the experiments including data processing, method, and comparisons, would be available for further verification, benchmarking, and development.

3) The work would benefit from verification on another body site but I can see that this might be out of scope for this manuscript; it would be informative, however, to discuss in more length the possible limitations and challenges that one might encounter when extending the analysis to other body sites.

* Minor

All data are said to be fully available without restriction. But this is human data. Does this mean that also detailed personal subject data is available without restriction? Does this respect and guarantee anonymity of the volunteers to a sufficient extent?

The figure quality can be substantially improved but I trust this is done in the next rounds.

60: Kim et al. (year)?

66: Veilonella -> Veillonella?

123: "S1 Table" -> Table S1 (?)

233: "analogous to PICRUST" -> why not just state the name of this method itself? I am not sure if we should take any specific toolkit as a standard in this field

391: remove the word "important" and let the readers evaluate the importance

Reviewer #2: The authors of this manuscript have made an extensive literature review to identify bacterial species and corresponding genes in the human skin microbiota that can be associated to aging. Molecular processes in human cells associated with the corresponding functionalities are also identified.

A bioinformatic analysis follows to identify groups of orthologous genes among the selected genes. These sequences are then used to build protein signatures (Hidden Markov Models). A sample collection obtained from cheek skin from younger and older individuals is used to evaluate the performance of the potential of this signatures to characterize processes associated to aging. A gut microbiome dataset is also used to test the specificity of these signatures.

The manuscript is interesting and combines a review of existing work on skin microbiome associated to skin aging with the analysis of a de novo acquired data set. However, some of the results need to be changed or better explained as they currently don’t support all the presented conclusions.

Comments:

The manuscript is, to some extent exploiting. a circular argument: literature is mined to identify genes, bacterial genomes and host metabolic functions associated to skin aging. Then a data set of microbiome samples from old and young females is analysed to verify that this indeed the case.

While this approach has merit, it is difficult to see how this approach could be used to generate new hypothesis. This part in the discussion should be further expanded.

Hidden Markov Models: The authors present the generated collection of HMMs as a useful tool for further research. However this collection is not presented or described in any way (I was expecting to find some link to this collection, but I have not been able to find it). These should be presented and discussed.

Hidden Markov Models: The authors built a collection of bacterial genes (presented in table 1) through literature mining. From this collection of genes and the genome sequences of bacterial strains associated to skin aging they built Hidden Markov Models through multiple sequence alignments, ortholog search and the use of hmmer. While this seems a sensible approach I really don't understand why they don't use the already existing signatures that can be found in databases such as PFAM, Interpro, TIGRFAM and the like. The advantages of using these standardized signatures are obvious as they would provided a more consistent set of identifiers that would enable comparisons between and across datasets . Also the signatures on these databases have been built by considering a more extended collections of genomes should they could be of more use if different species/genera are to be analysed. The authors need to justify why they chose to disregard the already existing databases of standardized signatures.

Validation on skin derived and gut samples: Figure 2 compares the occurrence of these motifs in skin derived and gut derived microbiome samples. However the results don't clearly show these motifs to be more often present on the skin samples or in the gut samples. Analysis of Figure 2 shows that these motifs are also found in the gut microbiome. The authors should explain better to what extent Figure 2 represents a validation of the approach. Also the skin samples in Figure 2 don't seem to be specifically related to aging, so the question is to what extent they expect to have more occurrence of these motifs in these samples.

Figure 5 shows the link between the pathway level scores and the SA score however it seems that is only for the younger individuals. There seems to be some link, however more exhaustive anlaysis should be presented. One of the main questions is if there is a significant difference in these pathway level scores between individuals with low or high SA. Maybe a t-test or a correlation analysis would provide convincing evidence that would complement the plot on Figure 5. Also it is not clear why the analysis for the older individuals is not presented.

I find Appendix S2 to be very informative as it provides a nice review of literature findings. Table S1 provides the list of reference genomes used for this task. It would become more informative if the genes/functions reported in literature that made the authors select each genome were also be reported.

6. PLOS authors have the option to publish the peer review history of their article (what does this mean?). If published, this will include your full peer review and any attached files.

Reviewer #1: **Yes: **Leo Lahti

Reviewer #2: No

---

## [Author Response · Author response to Decision Letter 0]

3 Jun 2021

Pleae find our responses in the cover letter and response to reviewers files.

---

## [Decision Letter · Decision Letter 1]

11 Aug 2021

PONE-D-20-39781R1

Charting host-microbe co-metabolism in skin aging and application to metagenomics data

PLOS ONE

Dear Dr. Kortman,

Thank you for submitting your manuscript to PLOS ONE. After careful consideration, we feel that it has merit but does not fully meet PLOS ONE’s publication criteria as it currently stands. Therefore, we invite you to submit a revised version of the manuscript that addresses the points raised during the review process.

My apologies for the delayed decision, which is mainly due to the holiday period. Your manuscript has been evaluated by the same experts. As you can see in their comments, the manuscript was found significantly improved, but lacked important (background) information that hampered a proper detailed evaluation, such as the source code. Please check the PLOS ONE policy on code sharing and adapt accordingly in a revision: **https://journals.plos.org/plosone/s/materials-software-and-code-sharing#loc-sharing-code**.

We look forward to receiving your revised manuscript.

Kind regards,

Erwin G Zoetendal, PhD

Academic Editor

PLOS ONE

Reviewers' comments:

Reviewer's Responses to Questions

**Comments to the Author**

1. If the authors have adequately addressed your comments raised in a previous round of review and you feel that this manuscript is now acceptable for publication, you may indicate that here to bypass the “Comments to the Author” section, enter your conflict of interest statement in the “Confidential to Editor” section, and submit your "Accept" recommendation.

Reviewer #1: (No Response)

Reviewer #2: (No Response)

2. Is the manuscript technically sound, and do the data support the conclusions?

Reviewer #1: Partly

Reviewer #2: Yes

3. Has the statistical analysis been performed appropriately and rigorously? 

Reviewer #1: I Don't Know

Reviewer #2: Yes

4. Have the authors made all data underlying the findings in their manuscript fully available?

Reviewer #1: No

Reviewer #2: No

5. Is the manuscript presented in an intelligible fashion and written in standard English?

Reviewer #1: Yes

Reviewer #2: Yes

6. Review Comments to the Author

Reviewer #1: Source code for the experiments has not been provided. Sharing the source code is generally recommended as good practice in bioinformatics research. It is a common misconception that sharing source code equals to delivering a ready-made software product. These are two different concepts, source code for the experiments can be provided "as is", in the form that they were when performing the analyses. If important part of the analysis were conducted manually, and are not documented as source code, the reproducibility and replicability of the analyses is severely limited. The source code should be shared unless there are binding justifications for not doing so. The lack of source code prevents the further evaluation of the implementation reliability, accuracy, robustness, and reproducibility.

Reviewer #2: The authors have addressed all my comments. However an important part of my comment pertains the availability of additional data describing the HMMs and the input files. These files are mentioned in the description of the supporting information but are not available for download. Sometimes submission systems don't easily allow this type of supplementary files. If that is the case, I suggest the authors deposit them in some repository (such as gitlab or github) and include the link.

7. PLOS authors have the option to publish the peer review history of their article (what does this mean?). If published, this will include your full peer review and any attached files.

Reviewer #1: No

Reviewer #2: No

---

## [Author Response · Author response to Decision Letter 1]

17 Sep 2021

Please find our response in the files "Response to reviewers" and "Cover letter resubmission".

---

## [Decision Letter · Decision Letter 2]

11 Oct 2021

Charting host-microbe co-metabolism in skin aging and application to metagenomics data

PONE-D-20-39781R2

Dear Dr. Kortman,

We’re pleased to inform you that your manuscript has been judged scientifically suitable for publication and will be formally accepted for publication once it meets all outstanding technical requirements.

N.B. I noticed that the older age group is described as 59-65 under heading “Sample collection and phenotypic assessment” while it is described as 59-68 a few lines later under heading “Study design and skin swab collection”. Make sure to correct this difference in the proof.

Kind regards,

Erwin G Zoetendal, PhD

Academic Editor

PLOS ONE

Additional Editor Comments (optional):

Reviewers' comments:

Reviewer's Responses to Questions

**Comments to the Author**

1. If the authors have adequately addressed your comments raised in a previous round of review and you feel that this manuscript is now acceptable for publication, you may indicate that here to bypass the “Comments to the Author” section, enter your conflict of interest statement in the “Confidential to Editor” section, and submit your "Accept" recommendation.

Reviewer #2: All comments have been addressed

2. Is the manuscript technically sound, and do the data support the conclusions?

Reviewer #2: (No Response)

3. Has the statistical analysis been performed appropriately and rigorously? 

Reviewer #2: (No Response)

4. Have the authors made all data underlying the findings in their manuscript fully available?

Reviewer #2: Yes

5. Is the manuscript presented in an intelligible fashion and written in standard English?

Reviewer #2: (No Response)

6. Review Comments to the Author

Reviewer #2: (No Response)

7. PLOS authors have the option to publish the peer review history of their article (what does this mean?). If published, this will include your full peer review and any attached files.

Reviewer #2: No

---

## [Editor Report · Acceptance letter]

18 Oct 2021

PONE-D-20-39781R2 

Charting host-microbe co-metabolism in skin aging and application to metagenomics data 

Dear Dr. Kortman:

I'm pleased to inform you that your manuscript has been deemed suitable for publication in PLOS ONE. Congratulations! Your manuscript is now with our production department. 

Kind regards, 

on behalf of

Dr. Erwin G Zoetendal 

Academic Editor

PLOS ONE